# Examination of the Corrosion Behavior of Shape Memory NiTi Material for Biomedical Applications

**DOI:** 10.3390/ma16113951

**Published:** 2023-05-25

**Authors:** Aboujaila A. M. Soltan, İsmail Esen, Seyit Ali Kara, Hayrettin Ahlatçı

**Affiliations:** 1Mechanical Engineering, Engineering Faculty, Karabuk University, Karabuk 78050, Turkey; seyitalikara.engineer@gmail.com (A.A.M.S.); iesen@karabuk.edu.tr (İ.E.); 2Material and Metallurgy Metallurgy Engineering, Engineering Faculty, Karabuk University, Karabuk 78050, Turkey; hahlatci@karabuk.edu.tr

**Keywords:** memory alloy, nitinol, corrosion, wear, microstructure

## Abstract

In this study, corrosion and wear tests of NiTi alloy (Ni 55%–Ti 45%) samples, known as shape memory alloy, which offer a shape recovery memory effect between memory temperatures ranging from 25 to 35 °C, have been carried out. The standard metallographically prepared samples’ microstructure images were obtained using an optical microscope device and SEM with an EDS analyzer. For the corrosion test, the samples are immersed with a net into the beaker of synthetic body fluid, whose contact with the standard air is cut off. Electrochemical corrosion analyses were performed after potentiodynamic testing in synthetic body fluid and at room temperature. The wear tests of the investigated NiTi superalloy were carried out by performing reciprocal wear tests under 20 N and 40 N loads in a dry environment and body fluid. During wear, a 100CR6-quality steel ball of the counter material was rubbed on the sample surface for a total of 300 m with a unit line length of 13 mm and a sliding speed of 0.04 m/s. As a result of both the potentiodynamic polarization and immersion corrosion tests in the body fluid, an average of 50% thickness reduction in the samples was observed in proportion to the change in the corrosion current values. In addition, the weight loss of the samples in corrosive wear is 20% less than that in dry wear. This can be attributed to the protective effect of the oxide film on the surface at high loads and the effect of reducing the friction coefficient of the body fluid.

## 1. Introduction

The ability of a plastically deformed material (alloy) to return to its original shape by applying heat, the thermomechanical or magnetic load, is defined as shape memory [1,2,3]. NiTi alloys, which show shape memory alloy (SMA) characteristics, gain importance due to their excellent shape memory effect (SME), super flexibility, and biocompatibility. Today, 90% of commercial applications of SMAs use alloys such as binary NiTi or ternary NiTiNb, NiTiTa, and NiTiCu [4]. NiTi-based alloys have been widely used recently in medical implants, aerospace engineering, and industrial fields [4]. Moreover, due to the good wear, corrosion resistance, and biocompatibility properties of NiTi SMAs, these materials have become attractive for various medical applications such as surgical instruments, self-expanding cardiovascular and urological stents, bone fracture cation plates, and nails [5,6,7]. The adjustable hardness of NiTi alloys has applications in bone implants and regenerative medicine. The corrosion resistance of NiTi becomes extremely important for applications in the human body, as the amount and toxicity of corrosion products control alloy biocompatibility. Few studies of the corrosion resistance of NiTi in simulated human body fluids have been reported [4,8]. The applicability of NiTi alloy in biomedicine was examined in a review by Castelmann and Motzkin [9], and it was concluded that the biocompatibility properties of this alloy are good enough to continue studying possible applications. Regarding the corrosion properties, they reported a study by Buehler and Wang stating that the corrosion behavior of NiTi alloy should correspond to that of titanium, whose excellent corrosion resistance is well documented [10,11]. The corrosion resistance properties of NiTi SMA, like those of other non-noble implant metals, are based on the presence of a passive film on the surface. If the passive film is intact, a general corrosion problem is not anticipated, but it should be noted that some ion release also occurs in the passive state. Moreover, deterioration of the passive film can occur both during surgery and during its service life. Metal ions arising from these processes may cause problems in the organism; allergic reactions to Ni_2_ should be considered [12,13,14]. Any improvement in corrosion resistance achieved through surface modification will benefit the patient and long-term implant stability [15]. However, it should be noted that mechanical processes such as abrasion frequently increase the corrosion kinetics of implant materials. It would be advantageous if surface modification simultaneously improves corrosion resistance [16,17,18]. It is well documented by different authors that some NiTi shape memory alloys show higher wear resistance when compared with some steels and Ni- and Co-based alloys [19,20,21,22,23,24,25]. Most relevant studies have been conducted by Li, who analyzed wear behavior under various wear processes, along with tribological applications [26,27]. Jin and Wang, who examined the wear resistance of these alloys, concluded that NiTi SMAs are more resistant, especially under high load, compared with 38CoCrMoAl and Co45 alloys at high hardenability [28,29].

### The Novelty of the Study

Although NiTi SMAs are an important biomaterial, immersion corrosion tests in body fluid and wear behavior in dry and body fluids have not been studied sufficiently in the literature. In this study, the dry and corrosive wear behavior, immersion and potentiodynamic polarization corrosion behavior in body fluid, and failure mechanisms of NiTi SMAs were examined in detail, and the results are reported.

However, few studies have shown a clear relationship between tribological properties such as abrasion resistance and sediments, Ms temperature, and environmental conditions. This indicates that further experimental work is required to understand the properties mentioned above in NiTi SMAs.

## 2. Materials and Methods

### 2.1. Microstructure Test

For the microstructure test of the studied NiTi superalloy, a sample of 13 × 10 × 0.3 mm was cut out from the 25.6 × 100 × 0.3 mm NiTi shape memory alloy sample firstly and then placed in a silicon mold and taken with Bakelite resin and hardening liquid. The surfaces of the Bakelite sample were sanded with 600, 800, 1200, 2000, and 2500 sandpaper in a metallographic machine and then polished with diamond water. The polished sample was immersed in an absolute solution of 1 mL of HF, 4 mL of nitric acid, and 5 mL of ethanol prepared in a container and etched for 90 s. After the acidified sample was removed from the solution and cleaned with alcohol, its microstructure was examined under a microscope and its photographs were taken with 10×, 20×, 50×, and 100× zoom and recorded.

### 2.2. X-ray Diffraction

Samples of appropriate sizes were cut to determine the chemical spectrum of the shape memory alloy and the texture of the phases. Then, their surfaces were prepared by standard metallographic procedures. An Ultima IV model XRD device was used in KBU Iron and Steel Institute. XRD graphics were taken in the 15–90° range in the XRD device, and the results were determined.

### 2.3. Corrosion Test

#### 2.3.1. Immersion Test

A sample of 13 × 10 × 0.3 mm was cut out from the NiTi shape memory alloy sample of 25.6 × 100 × 0.3 mm.

After the sample’s surface was polished with 2500 sandpaper, it was ultrasonically cleaned with alcohol. The initial weight was measured on an electronic balance with a precision of 0.1 mg and immersed in the body fluid medium. The sample was removed from the solution every 24 h, and the corrosion products on the sample surface were engaged in 50% H_2_SO_4_ + 50% HCl acid in an ultrasonic device for an average of 8 min and brushed with a soft brush cleaned with distilled water and alcohol before it was dried. The immersion corrosion tests (ASTM G31-21) were evaluated by measuring the weight losses and calculating changes in the thickness of the samples. In total, the test period was ten days.

#### 2.3.2. Potentiodynamic Polarization Test

For the potentiodynamic polarization tests (ASTM G215-17 and ASTM G220-20) of the examined NiTi super alloy, one side of the sample surface of 8 × 8 mm^2^ area was exposed to air, and the other surface was molded in conjunction with copper wire. In the test cell (Isotherm 3.3, 250 mL beaker), the test sample was placed as the working electrode, the graphite rod was identified as the counter electrode, and the saturated calomel electrode (SCE) was placed as the reference electrode. In all the experimental studies, especially without current carried through the system, the working and reference electrode was immersed in Hank’s solution (known as synthetic body fluid) at room temperature, and the variation of the corrosion potentials in millivolt (mV) between both of them were measured while the current was flowing. After reaching the equilibrium potential, the potentiodynamic polarization curves were created in the range of −0.25 v + 0.25 v at a scanning speed of 1 mv/s and recorded from the cathodic direction to the E_corr_ direction. Corrosion potential (E_corr_) and corrosion current density (I_corr_) were calculated by using the Tafel curves. For all parameters, three tests were performed on the Gamry Instruments potential dynamic polarization test device, and the arithmetic mean of the results was taken. This experiment continued for 23 days.

### 2.4. Wear Test

Wear tests of the NiTi SMAs were conducted by reciprocating wear tester (ASTM G133-22, manufacturer: Kett, FL, USA) under 20 N and 40 N loads in a dry environment and body fluid. During wear, a 100CR6-quality steel ball was taken with counter material on the sample surface with a 13 mm length and 0.04 m/s sliding speed, in total 300 m. The wear results were evaluated by weight loss measurement, conducted on an electronic precision balance with 0.1 mg sensitivity before and after the experiment. In addition, the width and durability of the track were measured by the wear family profilometer device to evaluate the wear losses as the loss of the created wear track area. It was examined using a scanning electron microscope to evaluate the mechanism. The test setup is shown in Figure 1.

## 3. Results

### 3.1. Mıcrostructural Results

The alloy optical microstructures examined at low magnification (10×) and high magnification (50×) are given in Figure 2a,b, respectively. Directional twinned martensite (detwinning martensite) was observed during the transition between the austenite and the martensite phase in this material structure; directional twinned martensite (detwinning martensite) was observed. Figure 2b shows the detwinning martensite structure in the rectangular region. This oriented twin martensite is considered the main deformation mechanism of superelastic deformation, which is compatible with the work of Wiley et al. [31]. When the microstructure is examined, it seems that there are many phases. However, it is thought that this is due to the condensation of titanium at specific points and nickel at other points [32,33]. In the studies of Kaya et al. [34], NI_4_Ti_3_ and austenite phases were found. Consistent with the studies of Kaya et al. [34], it was observed that the dark-colored regions were Ni_4_Ti_3_ phase at low temperatures, while the light-colored areas were found to represent the austenite phase. It is also seen in XRD analyses that these precipitates are Ni_4_Ti_3_, Ni_3_Ti, and NiTi_2_.

SEM images and EDS analysis results of the shape memory alloy are given in Figure 3 and Table 1. The surface formed by the overlapping of single traces is smooth and nonporous. In addition, the surface of the examined samples showed a white–gray appearance. It is believed that TiO_2_ or NiO_2_ is partially formed in regions 1, 3, 4, 5, 6, and 7, given in Figure 3b, and the NiTi matrix is found in region number 2.

#### XRD Analysis

According to the result of XRD (X-ray mass diffraction) analysis performed to determine the phases of the shape memory alloy, the NiTi austenite phase was detected, as seen in Figure 4. The shape memory state occurs in shape memory materials depending on the martensitic transformation. Nitinol B_2_ exhibits a stable structure in the austenite phase at room temperature. Morawiec et al. [35] reported that after cold rolling-induced deformation, NiTi SMA showed a negligible amount of martensitic transformation due to the high density of dislocations and clearance errors in the crystal structure of the material. In the microstructure results given in this study, it was observed that the twinned martensitic structure, austenite, martensite, and Ni_4_Ti_3_, Ni_3_Ti, NiTi, NiTi_2_, and Ti_2_Ni phases were formed. Similar phases were detected in the study by Kuang et al. [36].

### 3.2. Corrosion Test Results

#### 3.2.1. Immersion Corrosion Tests Results

Figure 5 gives the time (day)–weight loss (mg/dm^2^) relationship of the studied NiTi alloy. As seen in Figure 5, mass loss was almost negligible in the 0–2-day interval, and a linear mass loss was observed after the second day. The passive film was formed on the surface following the 8th day of the immersion test period, and there was a constant mass loss of the examined alloy.

SEM images and EDS analysis areas of the shape memory alloy are given in Figure 6. When the SEM images of the studied alloy in Figure 6 is examined, almost smooth and low-density wide-opening cavities are seen. Surface quality is very important for shape memory materials because it directly affects the formation of a passive structure. The study by Matthews et al. [37] examined the relationship between corrosion resistance and porosity. Porosity qualifies as a defect in the structure, because when naturally weak bonds are pulled on the metal surface, many porous structures can form on the metal surface. It is very difficult to form a passive film on the surface of a porous material. It is possible to form a continuous and smooth passive film on a clean surface, and in this study, such a surface appears, as shown in Figure 6. This is in line with good corrosion behavior. This oxide layer provided very good protection for the alloy under study and improved corrosion resistance. When the EDX results were analyzed (Table 2), titanium levels were high. Therefore, it is thought that TiO_2_ is formed; the CO_2_ layer in regions 2 and 7 and the high amount of Cu in regions 3 and 4 indicate precipitate formation.

The EDS analysis results for the regions 1–7 in Figure 6c is presented in Table 2.

#### 3.2.2. Potentiodynamic Corrosion Tests Results

The potentiodynamic curves (Tafel curves) of the investigated NiTi SMAs obtained at certain intervals in the Hanks solution up to 23 days are shown in Figure 7. Corrosion current density (I_corr_) and corrosion potential (E_corr_) were found from the extrapolation (Figure 7) of the intersection point in anodic and cathodic polarization. The changes in I_corr_ and E_corr_ over time are given in Figure 8. As see in Figure 8a, the corrosion current density changes periodically repetitive oscillation in the range of 50.10^−9^ to 490.10^−9^ A/cm^2^. The corrosion current density of the examined SMAs decreased to a minimum at 2 days and peaked on about the 15th day. While it was 50.10^−9^ A/cm^2^ on the 2nd day, it reached the 490.10^−9^ value again on the 15th. The corrosion current density was 280 × 10^−9^ A/cm^2^ on average. Figure 8b shows that the corrosion potential values shift in the most noble direction, showing resistance to corrosion in the solution versus time. In minimizing the corrosion current, the corrosion potential remains constant. It is seen that the corrosion potential peaks in the noble direction at the point where the corrosion current shows its maximum, and the corrosion potential remains constant on the days when the corrosion value decreases again.

A wide presence of passivation is observed for NiTi. In addition, it is believed that the reason for the increase in the current density of the NiTi alloy at high potentials on the 15th day is the formation of oxygen, corresponding to the oxidation of water. NiTi gains good corrosion resistance with the formation of the surface oxide film at this point. This is thought to be due to the formation of a protective oxide film of titanium oxide. These results are reported to show the tendency to form a passive film, with such data already reported by some authors [38,39,40].

According to Faraday’s rules, corrosion rate values leading to a reduction in the thickness in mm/year were calculated by using the corrosion current density (Figure 8a) and weight loss values (Figure 5), which are given in Figure 9. The reduction in thickness (Figure 9a) calculated with the potentiodynamic polarization data shows a repetitive variation in the range of 0.008–0.0008 mm/year over time. However, the changes in thickness calculated using the data from the immersion tests decreased from 0.06 to 0.028 mm/year over time, as presented in Figure 9b. Since the average test thickness of the potentiodynamically tested sample changes, the annual rate is also variable. The decrease in thickness changes and the average value is calculated as 0.005 mm/year. In immersion corrosion tests, the loss in thickness is continuously decreasing.

Although Cisse et al. (2022) applied chemical passivation coating on NiTi, the corrosion rate (0.11 × 10^−3^ mm/year) determined in this study is higher than the corrosion rate exhibited by the NiTi sample they examined [41]. In a study by E. Balcı et al. (2021), the average corrosion rate of NiTiNbV quaternary SCAs at different rates was calculated as approximately 1.70 × 10^−2^ mm/year, which is higher than the corrosion rate in this research [42]. It was determined that the corrosion resistance values in the potentiodynamic and immersion test performed in artificial body fluid at room temperature were higher than those obtained by other studies [41,42].

SEM images and EDS analysis results of the shape memory alloy are given in Figure 10. The number of pits on the corroded surface of the examined alloy, which is given in Figure 10a, is low. The existing pits are also so small that they cannot be observed, even at the high magnifications of 100× in Figure 10b and 500× in Figure 10c. In addition, passive films formed on the corrosion surface are compatible with the EDX result in Table 3 and Table 4. According to EDX analysis, it is seen that oxygen and titanium are generally high. It is generally believed that the TiO_2_ and CO_2_ layer is formed for this reason.

The EDS analysis results of the regions marked by 1–7 in Figure 10a are given in Table 3.

### 3.3. Wear Test Results

Figure 11 gives the variation of wear test results of The NiTi SMAs with applied load in a dry environment and body fluid. While the weight loss of the investigated NiTi SMAs in the dry environment was slightly lower than the weight loss in the body fluid at low loads, they exhibited quite a lot of weight loss at high loads. Figure 12 depicts the change in wear rates during the wear test of NiTi shape memory alloy with applied load. The fact that the body fluid wear rate is 20% lower than the dry environment wear rate can be attributed to the fact that body fluid reduces friction, structural transformations during wear, and the formation of an oxide film on the surface.

According to the weight loss results, two different behaviors can be distinguished for shape memory alloys reported in the literature. These are grouped by phase. In β-phase TiNi alloys, the wear resistance is consistent with the hardness values of each alloy as the hardness increases. In such alloys, the high hardness increase with Ni content is associated with the presence of Ti_3_Ni_4_ precipitates, which are semiconformable in the B_2_ matrix of both the β-phase (2) and the β-phase (3) [41]. This is attributed to the contact surface and the ease or difficulty of each material absorbing energy during the wear test. The alloy under study offers a high transformation capacity because the phase transformation temperature is close to the test temperature. As a result, it absorbs more energy to restore its structure and, more significantly, inhibits different damage mechanisms. As a result, we can say that it exhibits good wear resistance.

In Figure 13 and Table 5, SEM and EDX images of the examined alloy are given after the wear test in a dry environment. Abrasive slip traces were formed on the worn surface of the alloy in the direction of the slip. Observation of the abrasive wear mechanism limited the output of the Ni matrix, which eroded towards the contact boundary with plastic deformation, resulting in further weight loss. When the SEM analysis is examined, a smooth and clean surface is seen due to the particle detachment from the sample surface due to the plastic deformation on the contact surface. When SEM and EDX analyses are examined, it is seen that TiO_2_ formation occurs in regions numbered 1, 5, 6, and 7, and the CO_2_ layer occurs in regions numbered 2, 3, and 4.

The EDS analysis results of the regions marked by 1–7 in Figure 13d are given in Table 5.

SEM images and EDS analysis results of the worn shape memory alloy in the body fluid are given in Figure 14 and Table 6. The literature reports that X-ray diffraction of the material after wear tests on Hank’s solution leads to nucleation and reorientation of martensite plates in β and martensite alloys, respectively [42]. In contrast to those presented under the dry wear test of the alloy under study, the plates appear to be more strongly reoriented after a phase transformation to the martensite phase under applied stress during testing [43]. This and the oxide layer formation improved the corrosive wear resistance compared with the dry wear resistance. When the SEM images are examined, it is seen that the analyzed alloy exhibits abrasive wear. According to the EDX analysis, it is thought that CO_2_ is formed in regions 2, 3, 5, and 7, and TiO_2_ is formed in regions 1, 4, and 6.

## 4. Conclusions

In this study, the dry wear behavior of nitinol, wear in Hank fluid, immersion corrosion behavior in body fluid and potentiodynamic polarization test, and microstructure characterization and wear mechanisms were examined in detail, and the following results were observed.

According to the 8-day immersion corrosion test results, the mass loss was very slow in the first two days: 0.002 at the end of the first day and 0.02 mg/dm^2^ at the end of the second day, accelerating. At the end of the 3rd day, it accelerated to 26, and the loss remained constant for 4 days. It increased to 31.13 on the 5th day, 41.43 on the 6th and 7th days, remained stable, and decreased to 39.89 mg/dm^2^ on the 8th day. The reason for this nonlinear behavior was evaluated as the breaking and reformation of the protective oxide layer on the surface.

While the mass loss was 0.0059 due to the 300 m dry abrasion test under 20 N load, the mass loss increased to 0.0075 g in the abrasion test in Hank fluid at the same load and distance. Under a load of 40 N, the mass loss was 0.0211 g in dry wear, while this value decreased to 0.01739 g in the wear test in Hank fluid. Wear rates were 1.7 and 1.35 (g/Nm) × 10^−9^ in dry and hank fluid wear tests, respectively.

According to the SEM examination results of the wear surfaces, the wear mechanism is mostly abrasive. However, TiO_2_ oxide layers on the wear surface were also observed.

I_corr_ readings in the 23-day electropotentiodynamic corrosion test were initially 471.7 × 10^−9^ (A/cm^2^) but fell to 47.03 × 10^−9^ (A/cm^2^) on the second day. After the second day, the increased I_corr_ value began to grow and peaked on the 15th day, rising to 499.33 × 10^−9^ (A/cm^2^) before decreasing to 273.33 × 10^−9^ (A/cm^2^) on the 23rd day. The cause for this change is the creation of a protective titanium oxide film on the surface at the start, and as the thickness of this film increases, the electrical resistance and current decrease. After a while, however, the surface coating reforms and the I_corr_ recent drops. This demonstrates that NiTi material can create a protective surface film even when subjected to high electric potential.

## Figures and Tables

**Figure 1 materials-16-03951-f001:**
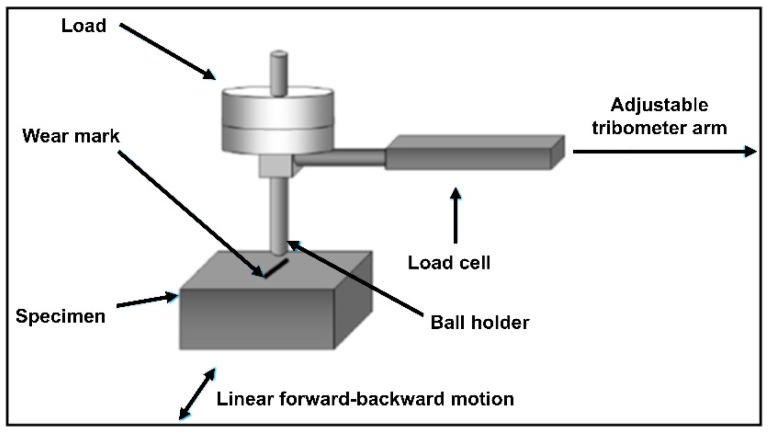
Schematic for the reciprocating abrasion wear test [30].

**Figure 2 materials-16-03951-f002:**
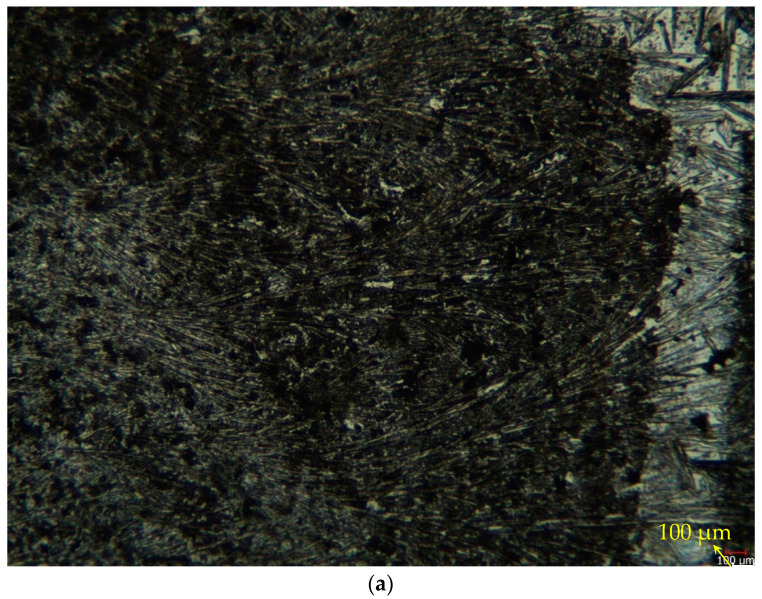
(**a**) Low-magnification (10×) and (**b**) high-magnification (50×) optical microstructure images of the alloy being studied. Read the arrow; the geometric drawing inside shows the detwinning martensite zones.

**Figure 3 materials-16-03951-f003:**
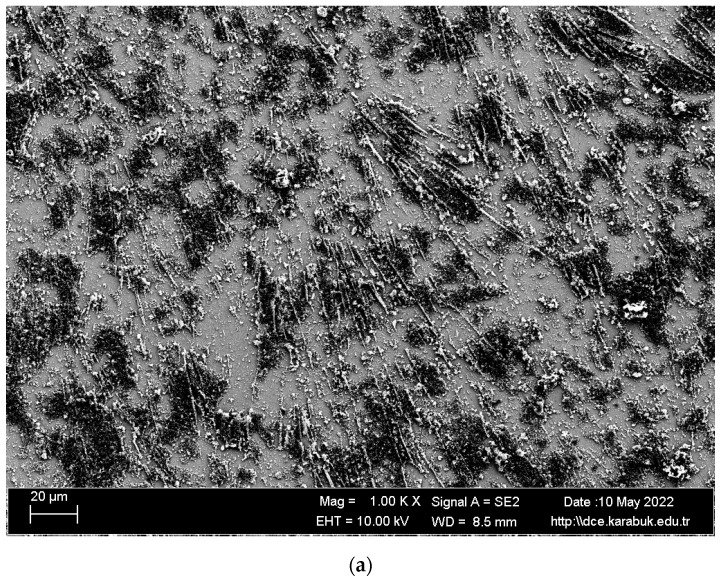
(**a**) Microstructure SEM images and (**b**) EDS analysis of shape memory alloy.

**Figure 4 materials-16-03951-f004:**
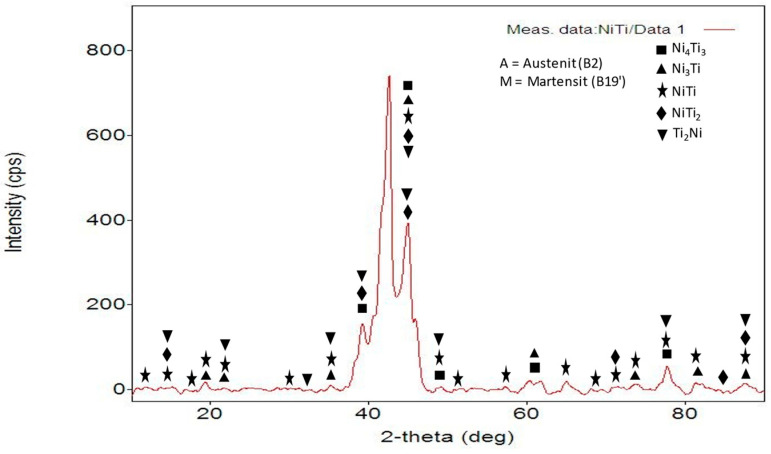
XRD analysis result of UNS shape memory alloy.

**Figure 5 materials-16-03951-f005:**
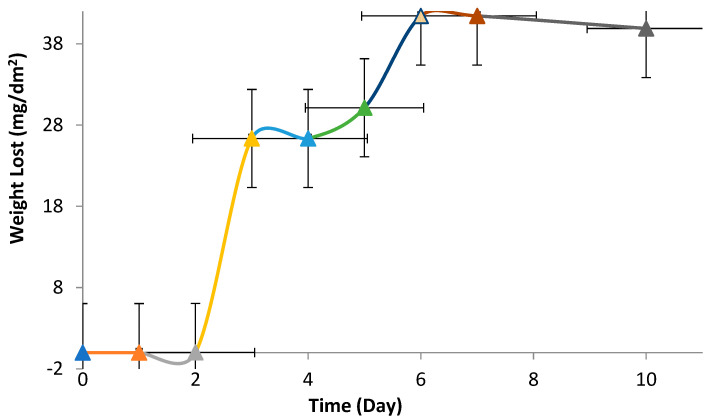
The time (day)–weight loss (mg/dm^2^) relationship of the examined NiTi alloy.

**Figure 6 materials-16-03951-f006:**
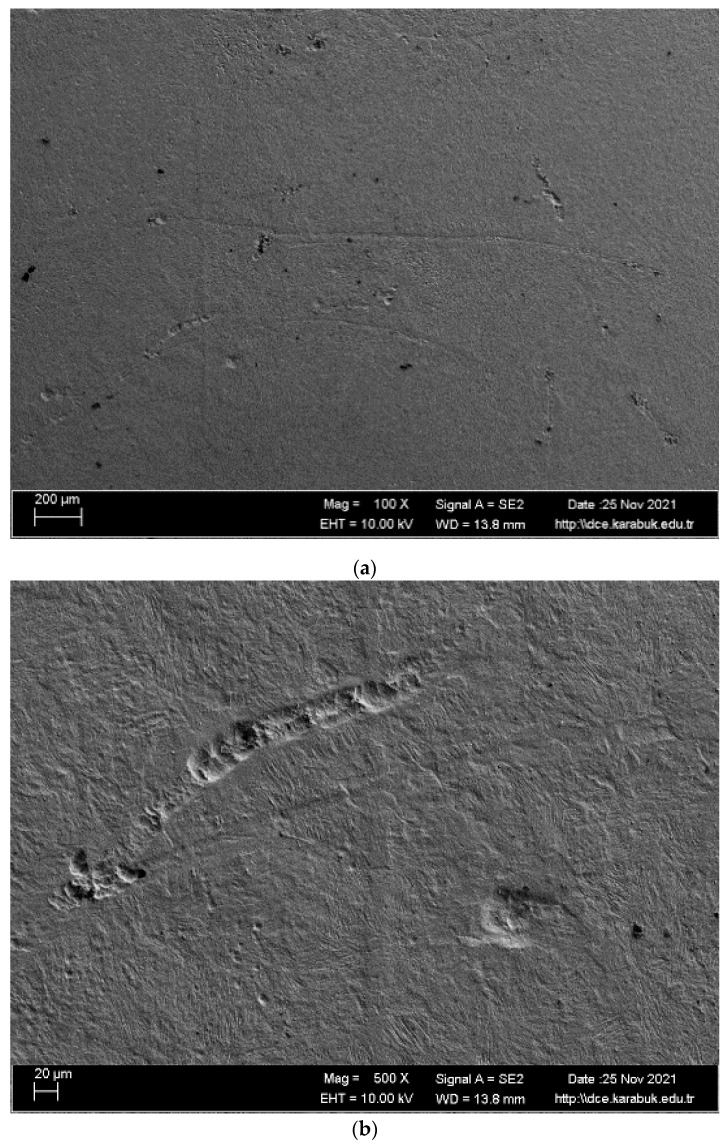
(**a**) and (**b**) SEM images and (**c**) EDS analysis areas of the NiTi SMAs after corrosion test.

**Figure 7 materials-16-03951-f007:**
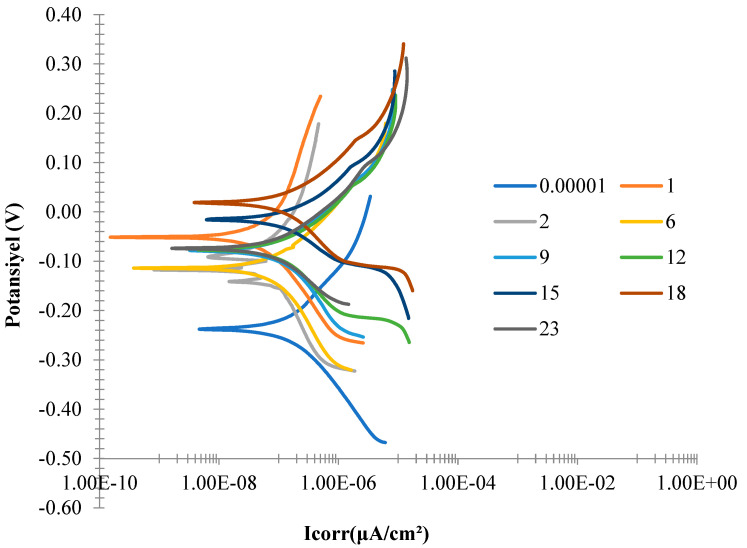
Taffel curves of the alloy were obtained in artificial body fluid for 23 days at room temperature.

**Figure 8 materials-16-03951-f008:**
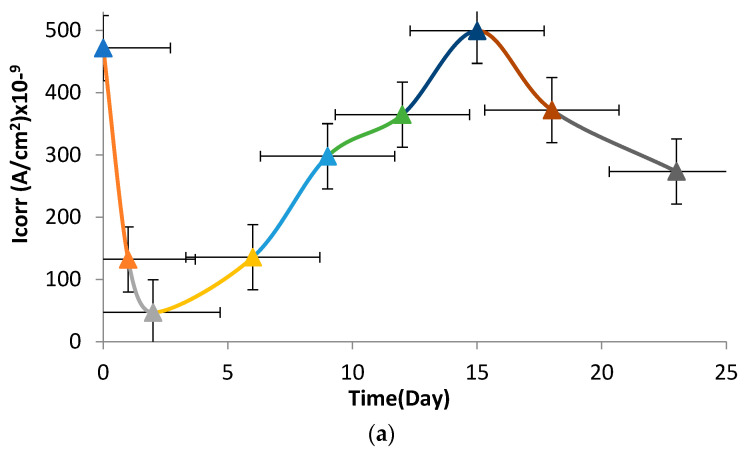
Changes in (**a**) corrosion current density (I_corr_) and (**b**) corrosion potential (E_corr_) of the NiTi SMAs over time in artificial body fluid at room temperature.

**Figure 9 materials-16-03951-f009:**
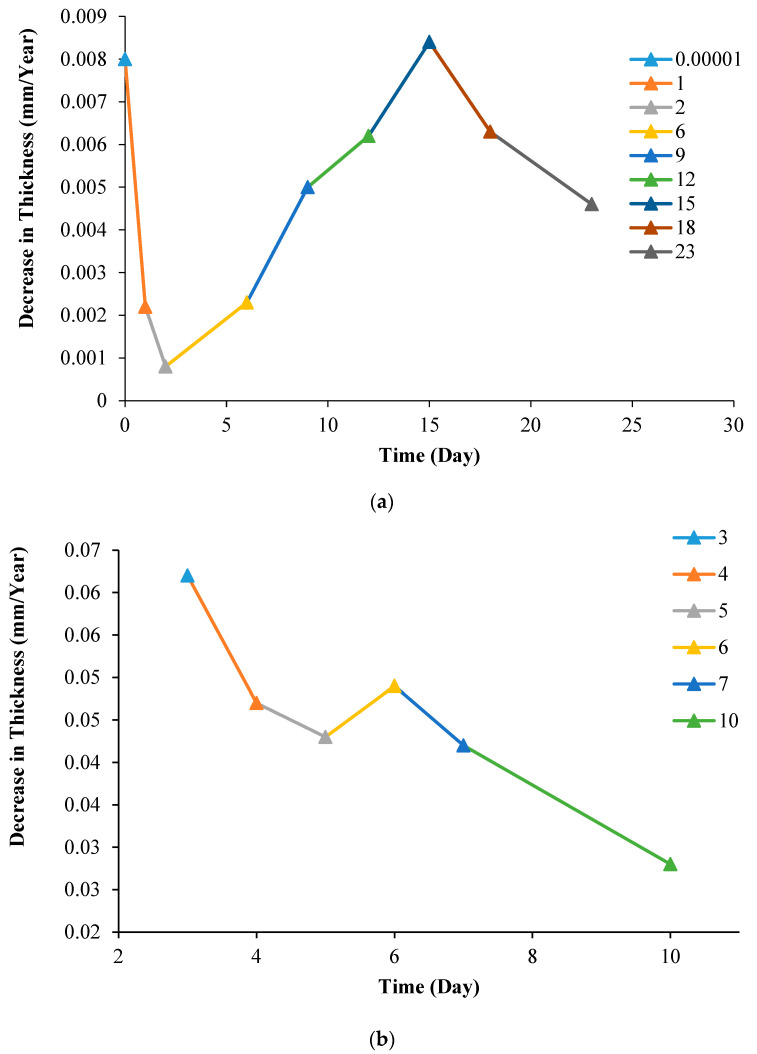
Decrease in thickness–time graph in mm/year after (**a**) potentiodynamic and (**b**) immersion test.

**Figure 10 materials-16-03951-f010:**
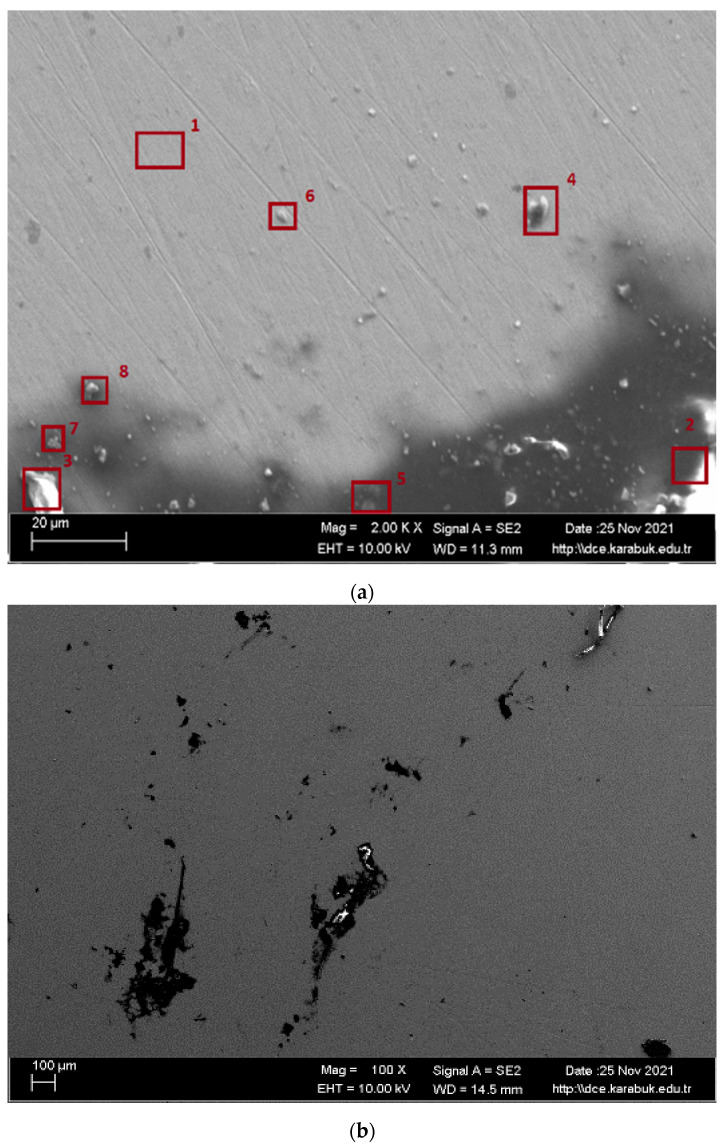
Postcorrosion SEM images: (**a**) at 2 k×; (**b**) at 200×; (**c**) at 500×.

**Figure 11 materials-16-03951-f011:**
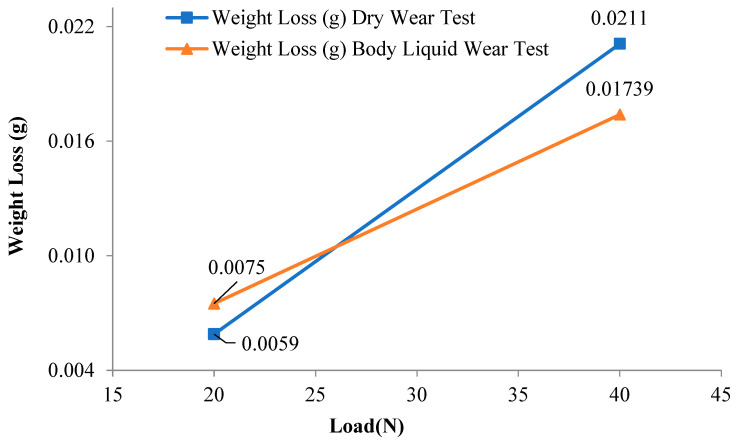
Variation of the weight loss (g) in dry and body fluid wear tests.

**Figure 12 materials-16-03951-f012:**
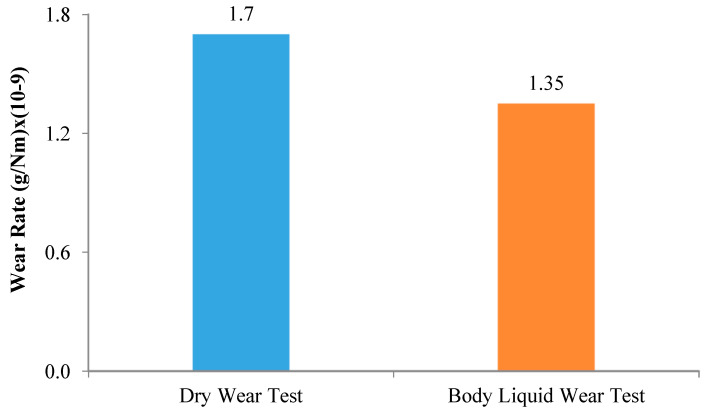
Variation in the wear rates of the NiTi shape memory alloy.

**Figure 13 materials-16-03951-f013:**
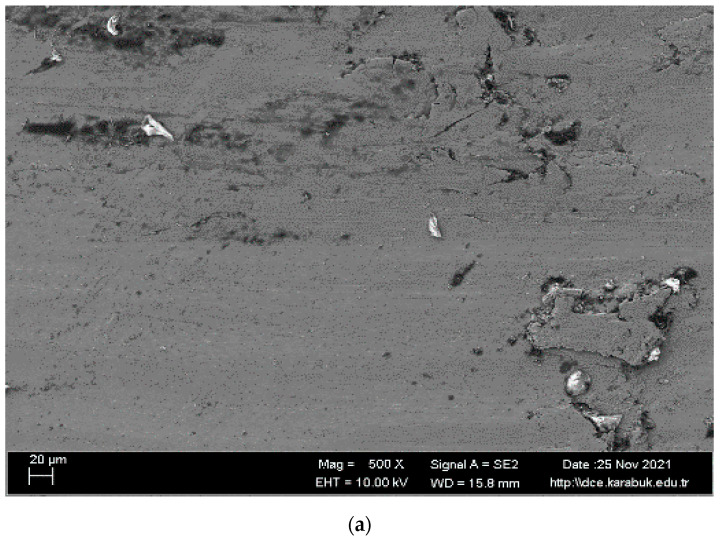
Worn surface SEM images of the shape memory alloy; (**a**) at 500×; (**b**) at 100×; (**c**) at 62× (**d**) at 2 k× after the dry wear test.

**Figure 14 materials-16-03951-f014:**
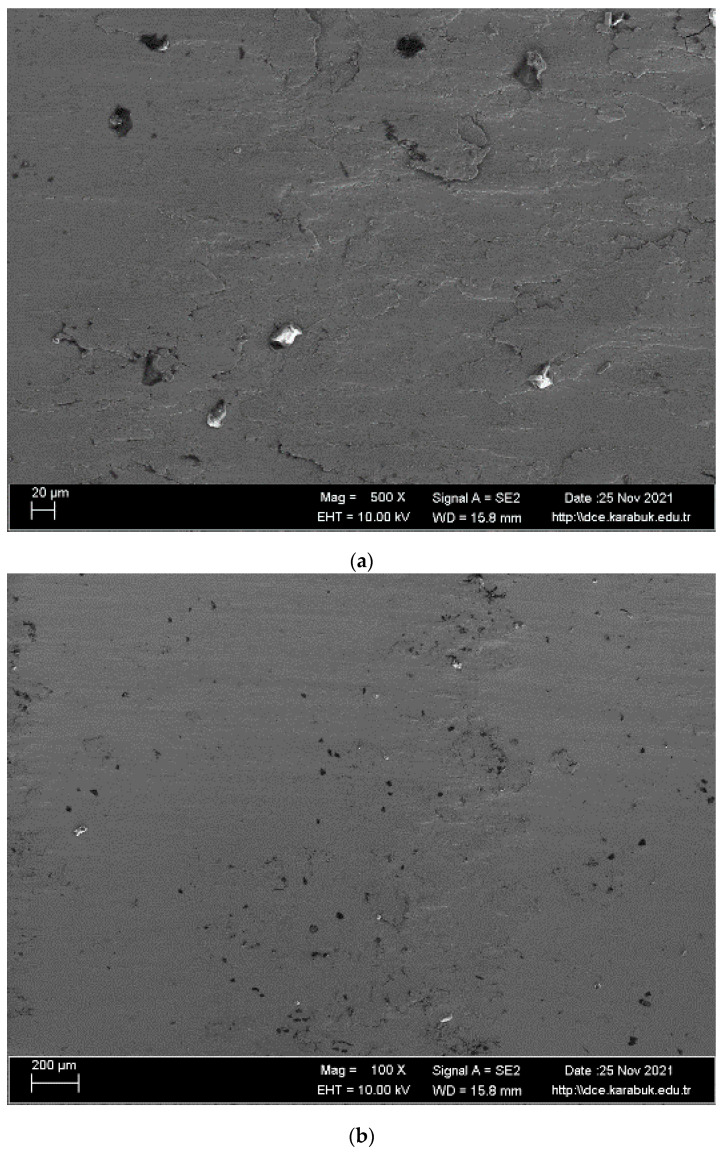
Worn surface SEM images of the shape memory alloy; (**a**) at 500×; (**b**) at 100×; (**c)** at 62× (**d**) at 2 k× after body fluid wear test.

**Table 1 materials-16-03951-t001:** The result of EDS analysis of shape memory alloy.

Mass Percent (%)											
Spectrum	H	C	O	Na	Mg	Cl	K	Ca	Ti	Ni	Cu
1	0.03	2.15	48.57	0.33	0.39	0	0.13	0.04	20.92	27.35	0.09
2	0	0.02	3.83	0.28	0.08	0.07	0.16	0	39.38	56.17	0
3	0.09	9.3	48.99	0.7	0.69	0.09	0	0.07	11.27	27.94	0.87
4	0.12	1.69	61.92	0.31	0.35	0	0	0	17.37	18.24	0
5	0.01	3.03	44.5	0.25	0.12	0.19	0.01	0	20.45	31.44	0
6	0.02	3	41.71	0.49	0.32	0.04	0	0	22	32.14	0.28
7	0.05	5.49	59.33	0.36	0.39	0	0.01	0.08	15.24	18.81	0.24
Mean value:	0.04	3.53	44.12	0.39	0.33	0.06	0.05	0.03	20.95	30.3	0.21
Sigma:	0.04	3.03	19.24	0.16	0.2	0.07	0.07	0.04	8.95	12.69	0.32
Sigma means:	0.02	1.15	7.27	0.06	0.08	0.03	0.03	0.01	3.38	4.79	0.12

**Table 2 materials-16-03951-t002:** The result of EDS analysis of shape memory alloy after corrosion test for the Figure 6c.

Mass Percent (%)									
Spectrum	C	O	Na	Mg	K	Ca	Ti	Ni	Cu
1	5.04	25.53	2.12	1.58	0.29	0.00	52.24	13.20	0.00
2	63.40	11.19	0.56	0.25	0.11	0.01	19.42	4.80	0.27
3	5.51	28.33	2.92	1.66	0.40	0.74	46.48	12.86	1.11
4	4.04	30.04	2.50	1.44	0.00	0.05	50.26	10.42	1.26
5	2.30	27.40	2.80	2.50	0.00	0.12	51.24	13.64	0.00
6	5.54	25.09	1.85	0.73	0.21	0.82	51.96	13.45	0.35
7	11.74	26.93	1.26	0.30	0.20	1.11	48.46	10.01	0.00
Mean value:	13.94	24.93	2.00	1.21	0.17	0.41	45.72	11.20	0.43
Sigma:	22.01	6.29	0.86	0.82	0.15	0.47	11.78	3.18	0.54
Sigma means:	8.62	2.38	0.32	0.31	0.06	0.18	4.45	1.20	0.20

**Table 3 materials-16-03951-t003:** The result of the EDS analysis of the shape memory alloy at 2 k× is magnificent.

Mass Percent (%)					
Spectrum	C	O	Ti	Ni	Cu
1	13.78	34.90	41.61	9.71	0.00
2	82.10	17.06	0.86	0.00	0.00
3	78.67	20.51	0.66	0.00	0.16
4	44.29	26.30	23.97	5,44	0.00
5	49.01	15.68	28.28	7.04	0.00
6	74.36	19.85	5.43	0.34	0.02
7	60.28	30.62	7.99	1.08	0.03
Mean value:	57.50	23.56	15.54	3.37	0.03
Sigma:	24.16	7.23	15.86	3.98	0.06
Sigma mean:	9.13	2.73	6.00	1.50	0.02

**Table 4 materials-16-03951-t004:** The result of EDS analysis of the shape memory alloy at 500× magnificent.

Mass Percent (%)					
Spectrum	C	O	Ti	Ni	Cu
1	15.15	33.98	41.95	8.85	0.05
2	68.23	27.63	3.96	0.04	0.14
3	72.71	27.28	0.01	0.00	0.00
Mean value:	52.03	29.63	15.31	2.96	0.07
Sigma:	32.02	3.77	23.16	5.09	0.07
Sigma mean:	18.48	2.18	13.37	2.94	0.04

**Table 5 materials-16-03951-t005:** The result of the EDS analysis of the shape memory alloy at 2k× magnificent after the dry wear test.

Mass Percent (%)									
Spectrum	C	O	Na	Mg	K	Ca	Ti	Ni	Cu
1	10.54	32.38	2.57	1.88	0.56	0.49	41.13	10.45	0.00
2	54.64	38.93	4.58	0.47	0.42	0.65	0.11	0.20	0.00
3	58.67	33.75	4.46	0.85	0.91	0.58	0.51	0.07	0.20
4	31.03	10.95	54.85	1.42	0.67	0.34	0.45	0.00	0.29
5	43.14	39.46	2.80	1.15	0.68	0.70	11.85	0.08	0.14
6	15.93	33.73	2.67	1.02	0.67	0.57	36.51	8.77	0.14
7	24.56	37.82	3.76	0.85	0.26	0.06	26.87	5.82	0.00
Mean value:	34.07	32.43	10.82	1.09	0.59	0.48	16.78	3.63	0.11
Sigma:	18.68	9.88	19.44	0.46	0.21	0.22	17.88	4.62	0.11
Sigma means:	7.06	3.73	7.35	0.17	0.08	0.08	6.76	1.74	0.04

**Table 6 materials-16-03951-t006:** The result of the EDS analysis of the shape memory alloy at 2 k× magnificent.

Mass Percent (%)									
Spectrum	C	O	Na	Mg	K	Ca	Ti	Ni	Cu
1	6.97	26.22	3.44	1.75	0.00	0.65	48.33	12.63	0.00
2	32.51	52.42	4.29	0.90	1.40	1.14	7.11	0.24	0.00
3	37.36	44.98	3.74	0.67	1.16	0.69	10.54	0.79	0.06
4	9.19	41.51	2.44	0.95	0.13	0.17	34.57	10.09	0.96
5	36.21	47.42	3.85	0.63	0.92	0.76	8.77	1.44	0.00
6	8.92	38.48	2.77	1.83	0.00	0.56	36.02	10.55	0.87
7	38.65	35.63	1.12	0.70	0.00	0.61	22.50	0.79	0.00
Mean value:	24.26	40.95	3.09	1.06	0.52	0.66	23.98	5.22	0.27
Sigma:	15.01	8.58	1.08	0.51	0.62	0.29	16.07	5.56	0.44
Sigma means:	5.67	3.24	0.41	0.19	0.23	0.11	6.07	2.10	0.17

## Data Availability

The results and data of this study can be shared with other scientists upon request.

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
