# Peer review of "Examination of the Corrosion Behavior of Shape Memory NiTi Material for Biomedical Applications"

_materials, 2023, doi:10.3390/ma16113951_

Round 1

Reviewer 1 Report

Dear author

Please adhere to the notes attached to the report

Reviewer 2 Report

Dear Authors, your article contains numerous serious shortcomings that must be removed:

·    Abstract: “In order to examine the shape memory effect in the alloy, to calculate the transformation temperatures, phase transition temperatures, and kinetic parameters, in other words, enthalpy and entropy values, measurements were made at KBU Iron and Steel Institute with the DSC 28 (differential scanning calorimetry) method.” I did not find any results related to this part of the Abstract.

·        Introduction: “The applicability of NiTi alloy in biomedicine was examined in a review by Castelmann 51 and Motzkin [7]”. This statement should be supported by more recent references.

·    Introduction: “Metal ions arising from these processes may cause problems in the organism; 61 especially allergic reactions to Ni+2 should be considered [9].” This statement should be supported by more recent references.

·         2.3.1. Immersion test: “The sample was removed from the solution every 24 hours, and the corrosion products on the sample surface were immersed in 50% H2SO4 + 50% HCl acid in an ultrasonic device for an average of 8 minutes and brushed with a soft brush cleaned with distilled water and alcohol before it was dried.” Why was this procedure chosen? Is it recommended for biomaterials? Can you cite a standard that recommends this procedure?

·         2.3.2. Potentiodynamic polarization: 3.5% SBF solution? Is it correct? Does it mean Hank’s solution?

·         2.3.2. Potentiodynamic polarization: “After reaching the equilibrium potential, the potentiodynamic polarization curves were created in the range of -0.25V + 0.25V ...”  this range is too narrow to assess a passive behavior … it should be broader in the positive direction in order to determine the pitting potential, which is the most important potentiodynamic parameter for metals/alloys with passive behavior.

·         Figure 4 (a): the scale is not readable.

·         Figure 5 (b): the red numbers are not readable.

·         3.2.1. Immersion Corrosion test results: “The passive film was formed on the surface following the 8th day of the immersion test period”. If this statement is true, the tested material could not be used in implantology at all. The surface of the implants in the environment of the human body must be in a passive state. If this is not the case, it is a serious failure of the implant.

·         3.2.2. Potentiodynamic Corrosion Test Results: “A wide presence of passivation is observed for NiTi.” Which results are related to this statement? The curves are measured in very narrow range of potentials vs. open circuit potential. Neither passive current densities nor pitting potentials are determined or shown on the curves. Tafel analysis is not appropriate for metals with passive behavior when anodic dissolution is controlled by passive current density not by the corrosion current density (icorr).

·         Figure 10.: The picture is incomplete and unintelligible, the axes with scale are missing … Potential should be in V vs SCE.

 ·         4. Conclusions:  In my opinion, the conclusions regarding corrosion tests (namely 1. and 2.) are based on misinterpreted results. There are mentioned no conclusions related to wear tests.

·         References: 13 references is older than 25 years … It is not appropriate. 

Author Response

Response to Reviewer 2

Comment:

 Dear Authors, your article contains numerous serious shortcomings that must be removed:

  • Abstract: “In order to examine the shape memory effect in the alloy, to calculate the transformation temperatures, phase transition temperatures, and kinetic parameters, in other words, enthalpy and entropy values, measurements were made at KBU Iron and Steel Institute with the DSC 28 (differential scanning calorimetry) method.” I did not find any results related to this part of the Abstract.

Answer:

Since the aim and limit of this study is to examine the corrosion behavior of NiTi shape memory alloy and to present the results related to it, and in order not to prolong the article unnecessarily and to present the results of the current research, experimental results other than corrosion behavior are not given.

Comment:

Introduction: “The applicability of NiTi alloy in biomedicine was examined in a review by Castelmann 51 and Motzkin [7]”. This statement should be supported by more recent references.

  • Introduction: “Metal ions arising from these processes may cause problems in the organism; 61 especially allergic reactions to Ni+2 should be considered [9].” This statement should be supported by more recent references.

Answer:

Current references related to nickel allergy have been found and added as seen below.

“Metal ions arising from these processes may cause problems in the organism; allergic reactions to Ni+2 should be considered [10–12].”

Comment:

  • 2.3.1. Immersion test: “The sample was removed from the solution every 24 hours, and the corrosion products on the sample surface were immersed in 50% H2SO4 + 50% HCl acid in an ultrasonic device for an average of 8 minutes and brushed with a soft brush cleaned with distilled water and alcohol before it was dried.” Why was this procedure chosen? Is it recommended for biomaterials? Can you cite a standard that recommends this procedure?

Answer:

In this study, this medium: 50% H2SO4 + 50% HCl was preferred for immersion corrosion, since both immersion corrosion and potentio-dynamic polarization were performed. Other potential dynamic and corrosive wear tests were performed in Hank's fluid. This is because in addition to its behavior towards Hank's liquid, its behavior in a heavily acidic environment is to be given in addition.

Comment:

  • 2.3.2. Potentiodynamic polarization: 3.5% SBF solution? Is it correct? Does it mean Hank’s solution?

Answer:

There was a typo, the expression here indicates Hank's fluid, and has been corrected in the text as seen below:

“In all the experimental studies, especially without current carried through the system, the working and reference electrode was immersed in 3.5% Hank’s solution at room temperature, and the variation of the corrosion potentials in millivolt (MV) between both of them were measured while the current was flowing.”

Comment:

2.3.2. Potentiodynamic polarization: “After reaching the equilibrium potential, the potentiodynamic polarization curves were created in the range of -0.25V + 0.25V ...” this range is too narrow to assess a passive behavior… it should be broader in the positive direction in order to determine the pitting potential, which is the most important potentiodynamic parameter for metals/alloys with passive behavior.

Answer:

The referee is right in his opinion, but considering the current test conditions, test setup and the tested material, stable tests for -0.25V + 0.25V could be made at this stage and the results were presented.

Comment:

  • Figure 4 (a): the scale is not readable.

Answer:

In the figure given as the relevant figure 4(a) (old version) (2(a) in the new version), the scale information has been enlarged and made readable.

Comment:

Figure 5 (b): the red numbers are not readable.

Answer:

The numbers have been enlarged on figure 5 to be more visible (new version figure 3).

Comment:

3.2.1. Immersion Corrosion test results: “The passive film was formed on the surface following the 8th day of the immersion test period”. If this statement is true, the tested material could not be used in implantology at all. The surface of the implants in the environment of the human body must be in a passive state. If this is not the case, it is a serious failure of the implant.

Answer:

This is due to the highly acidic environment used in the immersion test, the purpose of this study was to examine the behavior of the NiTi alloy in implant applications..

Comment:

3.2.2. Potentiodynamic Corrosion Test Results: “A wide presence of passivation is observed for NiTi.” Which results are related to this statement? The curves are measured in very narrow range of potentials vs. open circuit potential. Neither passive current densities nor pitting potentials are determined or shown on the curves. Tafel analysis is not appropriate for metals with passive behavior when anodic dissolution is controlled by passive current density not by the corrosion current density (icorr).

Answer:

The referee is right in his opinion, but tests have been done and results are presented within the scope and limits of the current study, tests and investigations on the relevant subject are ongoing and new results will be reported after the analyzes are completed.

Comment:

  • Figure 10.: The picture is incomplete and unintelligible, the axes with scale are missing … Potential should be in V vs SCE.

Answer:

Figure 10 has been corrected.

Comment:

  1. Conclusions: In my opinion, the conclusions regarding corrosion tests (namely 1. and 2.) are based on misinterpreted results. There are mentioned no conclusions related to wear tests.

Answer:

Wear test results are added to the conclusions.

Comment:

References: 13 references is older than 25 years … It is not appropriate.

Answer:

Some outdated references were given due to the history of the topic, with an additional five new references added from the last five years.

Round 2

Reviewer 2 Report

I recommend to accept the article.